# ATF3-SLC7A7 Axis Regulates mTORC1 Signaling to Suppress Lipogenesis and Tumorigenesis in Hepatocellular Carcinoma

**DOI:** 10.3390/cells14040253

**Published:** 2025-02-11

**Authors:** Qinglin Zhang, Fengzhi Zhu, Yin Tong, Yunxing Huang, Jiangwen Zhang

**Affiliations:** 1School of Biological Sciences, The University of Hong Kong, Hong Kong SAR, China; u3008593@connect.hku.hk (Q.Z.); u3005790@connect.hku.hk (Y.H.); 2College of Food Science and Technology, Shanghai Ocean University, Shanghai 201306, China; m220301002@st.shou.edu.cn; 3Department of Pathology, School of Clinical Medicine, The University of Hong Kong, Queen Mary Hospital, Pokfulam, Hong Kong SAR, China; tongyin9@hku.hk; 4Centre for Oncology and Immunology, Hong Kong Science Park, Hong Kong SAR, China

**Keywords:** HCC, mTORC1 signaling, lipid synthesis, transcriptional regulation, enhancer

## Abstract

Hepatocellular carcinoma (HCC) poses a substantial global health burden, with poor prognosis and high mortality rates. Dysregulated lipid metabolism has emerged as a critical driver of HCC progression. While mTORC1 signaling is known to promote lipid synthesis in HCC, the regulatory mechanisms governing mTORC1 remain largely unclear. Here, we demonstrate that mTORC1 inhibition significantly reduces lipogenesis in HCC and uncover a regulatory axis involving the transcription factor ATF3 and the leucine–arginine transporter SLC7A7. Transcriptomic analysis of HCC patients reveals an inverse correlation between ATF3 expression and lipid synthesis, a finding corroborated by experimental validation. Mechanistically, ATF3 suppresses mTORC1 signaling, thereby inhibiting lipid biosynthesis, with SLC7A7 identified as a key intermediary in this process. Specifically, ATF3 binds to the enhancer region of SLC7A7, driving its transcriptional activation and subsequently restraining mTORC1 activity. Functional assays in ATF3-overexpressing and -knockdown HCC cell lines further confirm ATF3′s role as a tumor suppressor. Our study identifies a novel ATF3-SLC7A7-mTORC1 regulatory axis that attenuates lipogenesis and tumorigenesis in HCC, establishing a critical link between lipid metabolism and hepatocarcinogenesis. These findings offer new insights into potential therapeutic targets for the treatment of HCC.

## 1. Introduction

Hepatocellular carcinoma (HCC) is the most prevalent form of liver cancer, constituting approximately 90% of all liver cancer cases worldwide [1]. Despite advancements in therapeutic modalities such as transplantation and resection, HCC remains a formidable challenge, ranking sixth among the most diagnosed malignancies and fourth among cancer-related causes of mortality [2,3]. Notably, the incidence of HCC attributed to nonalcoholic fatty liver disease (NAFLD) has increased sharply in recent years, with NAFLD emerging as the fastest-growing etiology of HCC in western nations [3,4]. Central to NAFLD pathology is the excessive accumulation of triglycerides (TGs), primarily fueled by heightened de novo fatty acid (FA) synthesis within hepatic cells [5]. Consequently, aberrant upregulation of lipid synthesis emerges as a pivotal risk factor in NAFLD progression, with implications extending to HCC development [5,6]. A deeper exploration of the underlying mechanisms governing this process holds promise for enhancing our comprehension of HCC pathogenesis and may unveil novel therapeutic targets for HCC management.

Enhanced de novo lipid synthesis is a critical driver of HCC progression [7]. The process begins with the transformation from citrate to acetyl-CoA, which is subsequently converted into palmitic acid and monounsaturated FAs by ACLY, FASN, and SCD. FAs are then converted into TGs, which can be preserved as lipid droplets as an energy reserve to support the unchecked proliferation of cancer cells [8,9]. Elevated expression of ACLY, FASN, and SCD has been observed in HCC patients, indicating a link between increased lipogenesis and tumor development [10,11,12]. Importantly, activated mTORC1 signaling enhances lipid biosynthesis by promoting the stability and splicing of transcripts encoding ACLY, FASN, and SCD via activation of SRPK2 [13]. Moreover, mTORC1 stimulates the processing and transcription of SREBPs, key transcription factors that regulate FA synthesis [14,15]. Identifying novel pathways that modulate mTORC1 activity could offer new therapeutic opportunities to target dysregulated lipid metabolism in HCC.

Typical activators of mTORC1 include growth factors and amino acids such as arginine, leucine, and glutamine [13]. Transporters from the solute carrier family, which mediate the import of these amino acids, including SLC1A5 and SLC38A5, are upregulated in various cancers [16]. Therefore, further investigation of SLC family members, particularly those involved in amino acid transport, may reveal novel regulatory mechanisms governing mTORC1 signaling.

mTORC1 is also mediated by transcription factors as well as nutrients. Activating transcription factor 3 (ATF3) plays a multifaceted role in regulating immunity, metabolism, and tumorigenesis. Elimination of ATF3 has been shown to activate mTORC1-p70S6K signaling in liver inflammatory injury [17]. In the context of hepatocellular carcinoma (HCC), ATF3 predominantly functions as a tumor suppressor, with downregulated expression levels observed in HCC patients compared to healthy individuals [18,19]. Furthermore, beyond its involvement in oncogenesis, ATF3 has been implicated in mediating lipid anabolism, a process tightly associated with the progression of NAFLD and HCC. Previous studies have revealed that ATF3 binds to the promoter of ChREBP and represses its transcription, thereby inhibiting lipogenesis in mouse white adipocytes [20]. Despite these findings, the role of ATF3, the SLC family, and mTORC1 in lipid synthesis within the context of HCC remains elusive.

Here, we provide evidence that ATF3 inhibits lipid accumulation in HCC by modulating the SLC7A7-mTOR signaling axis, utilizing a combination of bioinformatics and experimental approaches. Mechanistically, we demonstrate that ATF3 directly binds to the enhancer region of SLC7A7, promoting its transcriptional activation and consequently suppressing mTOR signaling and lipid synthesis. Our findings elucidate a novel regulatory circuit involving mTORC1 in lipid metabolism and HCC tumorigenesis, establishing a crucial link between lipid homeostasis and cancer progression.

## 2. Materials and Methods

### 2.1. Cell Maintenance

The human HCC cell lines Huh7, PLC/PRF/5, and HepG2 and the human kidney cell line 293 T were acquired from the Japanese Collection of Research Bioresources (JCRB) Cell Bank, Osaka, Japan. Huh7, PLC/PRF/5, and 293 T cells were grown in high-glucose Dulbecco’s modified Eagle medium (DMEM, Gibco, New York, NY, USA), while HepG2 cells were cultured in William’s E Medium (Gibco). Both media were supplemented with 10% fetal bovine serum (FBS, qualified, Gibco, New York, NY, USA) and an antibiotic–antimycotic solution (Gibco). All cell lines were incubated at 37 °C in a humidified atmosphere of 5% CO_2_ using a Heracell VIOS 160i incubator (Thermo, Waltham, MA, USA).

### 2.2. RNA Extraction and Quantitative Real-Time PCR (qRT-PCR)

The assay was conducted as previously described [21]. Total RNA was isolated from the cell lines using RNAzol (Sigma, Burlington, MA, USA), and its concentration was measured with a NanoDrop 2000 spectrophotometer (Fisher, Pittsburgh, PA, USA). The isolated RNA was then reverse transcribed into cDNA using the Reverse Transcription Kit (Qiangen, Shanghai, China). Quantitative real-time PCR (qRT-PCR) was carried out on the ABI StepOne Plus system (Applied Biosystems, Foster City, CA, USA) using iTaq Universal SYBR Green Supermix (Bio-Rad, Hercules, CA, USA). The primer information is provided in Appendix A. Gene expression was normalized to the internal control genes GAPDH or RPL13A, and relative mRNA levels were calculated using the 2-ΔΔCT method.

### 2.3. Vector Construction

ATF3 coding sequence (CDS) was amplified from pRK-ATF3 (26115, Addgene, Watertown, MA, USA), and SLC7A7 CDS was amplified from Huh7 cDNA. Sequences of ATF3 promoter and SLC7A7 enhancer were amplified from Huh7 genome DNA, which was extracted and purified by using a Monarch Genomic DNA Purification Kit (New England Biolabs, NEB, Ipswich, MA, USA). These sequences were amplified by PCR. For construction of ATF3 and SLC7A7 expression vectors, CDS of ATF3 or SLC7A7 was inserted between the 2 Esp3I cutting sites of a modified lenti-3×Flag plasmid generated from lentiCas9-Blast (52962, Addgene, Watertown, MA, USA) using homemade hot fusion solution [22]. For construction of luciferase vectors, sequences of ATF3 promoter or SLC7A7 enhancer were inserted between the MluI and XhoI cutting sites of pGL3 promoter (Promega, Madison, WI, USA) with T4 ligase (NEB). ATF3 and SLC7A7 shRNA oligos were obtained from Integrated DNA Technologies (IDT, Hong Kong, China). Each of the oligos was annealed and inserted into the Esp3I cutting site of pLV-H1TetO-GFP-Bsd (Biosettia, San Diego, CA, USA). For DNA affinity assay, ATF3 cDNA was inserted into the BamHI cutting site of a modified pET-28a (+) vector (Novagen, Pasig, Philippines) with a SUMO tag to generate a recombinant ATF3 expression vector. Information of the primers and oligo utilized in cloning is provided in Appendix A.

### 2.4. Lentivirus Production and Infection

The 293 T cells were seeded in 60 mm plates (Corning, Shanghai, China) and co-transfected with lentiviral vectors for ATF3, SLC7A7, shATF3, or shSLC7A7, along with psPAX2 (Addgene, 12260, Watertown, MA, USA) and pMD2.G (Addgene, 12259, Watertown, MA, USA) using Lipo2000 (Vazyme, Shanghai, China). Viral supernatants were harvested at 48 and 72 h post-transfection and concentrated using PEG-it Virus Precipitation Solution (System Biosciences, Palo Alto, CA, USA). The concentrated virus was then used to infect target cells in the presence of 8 µg/mL polybrene. Twenty-four hours after infection, the cells were processed for subsequent experiments. To induce lentiviral vector expression, 1 µg/mL doxycycline was supplied to the culture media.

### 2.5. Transwell Assay

The assay was conducted as previously described [23]. First, 8 µm pore Transwell inserts were pre-coated with Geltrex (Gibco) and placed into 24-well plates (Corning, Shanghai, China). The lower chamber of each well was filled with 700 µL of DMEM supplemented with 10% FBS. Cells (1 × 10⁵) were resuspended in 300 µL of serum-free DMEM, introduced into the upper layer, and kept for 24 h. Non-migratory cells on the upper side of the membrane were erased using a cotton swab. Inserts were then collected, fixed with methanol for 10 min, and stained with crystal violet for another 10 min. Migrated cells were visualized and counted using an inverted microscope at 100× magnification.

### 2.6. Wound Healing Assay

The assay was conducted as previously described [23]. Cells were seeded into a 6-well plate (Corning). Once the confluency reached 80–100%, a scratch was manually created, and cells were washed twice with phosphate-buffered saline (PBS). Medium was changed into DMEM with 2% FBS and 1 mM thymidine. The scratch was photographed at 0 h, 24 h, and 48 h. The scratch area was measured by ImageJ v1.50d and the wound closure ratio was calculated.

### 2.7. Colony Formation Assay

The assay was conducted as previously described [23]. Cells were trypsinized, counted, and plated at a density of 2000 cells per well in 6-well plates (Corning). The cells were then cultured for 14–21 days to allow colony formation. Once colonies containing more than 50 cells became visible, the plates were processed by fixing the cells with 4% paraformaldehyde for 10 min, followed by staining with crystal violet for 10 min. The colonies were subsequently photographed and manually counted.

### 2.8. MTT Assay

The assay was conducted as previously described [23]. Three thousand cells were counted, resuspended in 100 µL of complete DMEM, and seeded into 96-well plates (Corning). The plates were maintained for 4–5 days, and cell proliferation was measured every day. Then, 10 µL of 3-[4,5-dimethylthiazol-2-yl]-2,5-diphenyltetrazolium bromide (MTT) was supplied into the wells, following with incubation at 37 °C for 3 h. The medium was removed, and 100 µL of DMSO was supplied into the wells to dissolve insoluble formazan. The plates were shaken for 15 min in the dark. Absorbance was then recorded at 490 nm.

### 2.9. Immunoblotting

The assay was conducted as previously described [21]. Cells were lysed in RIPA buffer for 30 min to obtain whole-cell lysates. Tumor tissues were similarly lysed in RIPA buffer and homogenized with a glass homogenizer to produce the lysate. The resulting lysate was centrifuged at 12,000 rcf for 3 min. Then, 5× Laemmli buffer was supplied to the supernatant, and the samples were kept at 95 °C for 20 min. The samples were then resolved with SDS-PAGE. The conditions were 80 V for 18 min followed by 140 V for 1 h. Proteins were transferred to 0.45 µm polyvinylidene difluoride membranes at 20 V for 40 min. Membranes were blocked with 5% bovine serum albumin for 30 min and then incubated with primary antibody overnight at 4 °C. After washing three times with Tris-buffered saline (TBS) containing 0.1% Tween 20 (TBST) for 5 min each, the membranes were exposed to a secondary antibody at room temperature for 1 h. Following another round of washing with TBST for 5 min 3 times, protein bands were visualized. Antibodies’ information is provided in Appendix A.

### 2.10. Confocal Assay

Cells were seeded into 12-well plates (Corning). Once the confluency reached 50–60%, cells were transfected with pcDNA3-TORCAR plasmid (Addgene, 64927, Watertown, MA, USA) and maintained for 24 h. After transfection, cells were photographed with a Carl Zeiss LSM 980 confocal laser scanning microscope (ZEISS, Oberkochen, Germany). The reporter activity was calculated by the intensity ratio of CFP/YFP.

### 2.11. Nile Red Staining Assay

Cells were seeded into 12-well plates (Corning, Shanghai, China). Once the confluency reached 50–60%, cells were washed twice with PBS and were fixed with 4% paraformaldehyde (PFA, Sigma, Burlington, MA, USA) for 10 min. After fixing, cells were washed three times with PBS. Nile Red (Invitrogen, Carlsbad, CA, USA) was dissolved in acetone with the final concentration of 1 mg/mL. Cells were stained with PBS supplied with 1 ug/mL Nile Red and 500 ng/mL Hoechst 33,342 solution (Dojindo Laboratories, Kanagawa, Japan) and were incubated on a shaker for 10 min in the dark. After incubation, cells were washed three times with PBS and were photographed with a BioTek Cytation1 Cell Imaging Multimode Reader. GFP fluorescence intensity was quantified using ImageJ v1.50d.

### 2.12. Oil Red O Staining Assay

Cells were seeded into 6-well plates (Corning). Once the confluency reached 70%, cells were washed twice with PBS and were fixed with 4% paraformaldehyde (PFA) for 10 min. After fixing, cells were washed twice with PBS. Oil Red O (Dieckmann, Shenzhen, China) was dissolved in 60% isopropanol with the final concentration of 3.5 mg/mL. Cells were stained with Oil Red O solution and incubated for 10 min. After incubation, cells were washed three times with PBS and were photographed using an inverted microscope.

### 2.13. Triglyceride Quantification Assay

Cells were seeded into 96-well plates or 10 cm plates (Corning). Once the confluency reached 90–100%, cells were subjected to a Triglyceride-Glo™ Assay Kit (Promega, Madison, WI, USA, for 96-well plate) or Triglyceride Assay Kit (Nanjing Jiancheng Bioengineering Institute, Nanjing, China, for 10 cm plate) to make samples. Luminescence or absorbance at 500 nm was measured using a BioTek Cytation1 Cell Imaging Multimode Reader.

### 2.14. Free Fatty Acid Quantification Assay

Cells were seeded into 10 cm plates (Corning). Once the confluency reached 90–100%, cells were subjected to a Free Fatty Acids (FFA) Content Assay Kit (Solarbio, Beijing, China) to make samples. Absorbance at 550 nm was measured using a BioTek Cytation1 Cell Imaging Multimode Reader (Winooski, VT, USA).

### 2.15. Chromatin Immunoprecipitation (ChIP) Assay

Huh7 cells that had been infected with ATF3 lentivirus were cultured in 15 cm plates (Corning). When the cells reached 90–100% confluency, they were washed twice with PBS and treated with DMEM containing 1% formaldehyde, followed by a 5 min incubation on a shaker. After this step, 900 µL of 2.5 M glycine was added, and the cells were incubated for an additional 5 min on a shaker. The cells were then washed twice with PBS, scraped off, and collected by centrifugation at 500 g for 5 min at 4 °C. They were resuspended in lysis buffer (85 mM KCl, 0.5% CA-630, 5 mM PIPES pH 8, 1 mM PMSF, and Pierce Protease and Phosphatase Inhibitor from Thermo) and sonicated for 150 s using the M220 Focused-ultrasonicator (Covaris, Woburn, MA, USA) at a peak power of 75 W, duty factor of 2%, 200 cycles/burst, and temperature maintained at 4 °C. Following sonication, the lysate was centrifuged at 1000 g for 5 min at 4 °C. The nuclear pellet was then resuspended in shearing buffer (10 mM Tris-HCl pH 8, 0.1% SDS, 1 mM EDTA pH 8, 1 mM PMSF, and Protease and Phosphatase Inhibitor) and subjected to sonication for 1560 s under the same conditions as before. After another centrifugation at 10,000 g for 10 min at 4 °C, the supernatant was collected. This supernatant was either stored as input or mixed with antibodies and incubated overnight on a shaker at 4 °C. After incubation, pre-blocked ChIP-grade Protein A/G Magnetic Beads (Thermo) were added and incubated on a shaker at 4 °C for 4 h. After this step, the beads were precipitated, and the supernatant was discarded. The beads were washed three times with sequential wash buffers: wash buffer I (150 mM NaCl, 20 mM Tris-HCl pH 8, 2 mM EDTA pH 8, 0.1% SDS, 1% Triton), wash buffer II (500 mM NaCl, 20 mM Tris-HCl pH 8, 2 mM EDTA pH 8, 0.1% SDS, 1% Triton), and wash buffer III (250 mM LiCl, 10 mM Tris-HCl pH 8, 1 mM EDTA pH 8, 1% sodium deoxycholate, 1% CA-630). The proteins were then eluted using elution buffer (1% SDS, 100 mM NaHCO_3_). All samples received a 5 M NaCl solution to achieve a final concentration of 0.2 M and were incubated overnight at 65 °C. After this incubation, the samples were treated with RNase A and Proteinase K, purified using the Monarch PCR & DNA Cleanup Kit (NEB), and subsequently analyzed by qRT-PCR.

### 2.16. Luciferase Reporter Assay

Huh7 cells infected with empty vector (EV) and ATF3 lentivirus were seeded into 12-well plates (Corning). Once the confluency reached 60–70%, cells were co-transfected with either pGL3-SLC7A7 enhancer or pGL3-ATF3 promoter plasmids and pRL-TK plasmid (Promega) and maintained for 24 h. After transfection, cells were washed twice with PBS and subjected to a Dual-Glo Luciferase Assay System Kit (Promega, Madison, WI, USA) to make samples. The luciferase activities were measured using a GloMax 20/20 Luminometer (Promega, Madison, WI, USA).

### 2.17. Mouse Xenograft Model

The assay was conducted as previously described [21]. This study was conducted following the ARRIVE guidelines, with the experimental protocol receiving approval from the Committee on the Use of Live Animals in Teaching and Research (CULATR) at the University of Hong Kong (CULATR No. 5712-21). License of animal experiments was approved by Hong Kong SAR Government, Department of Health. The Huh7 shATF3 cell line was suspended in sterilized PBS with a final density of 5 × 10⁷ cells/mL, and the cell suspension was injected subcutaneously into the flanks of 4- to 6-week-old female nude mice. Each mouse was inoculated with 5 × 10^6^ cells. Mice body weight and tumor volume were recorded every three days. Five weeks after inoculation, mice were anesthetized via intraperitoneal injection of 100 µL of 12.5 mg/mL tribromoethanol, followed by dissection for sacrifice, and the tumor xenografts were acquired.

### 2.18. Immunohistochemistry

Tumor tissues obtained from the mouse xenograft model were collected and fixed in 4% paraformaldehyde (PFA) for 48 h. Following fixation, the tissues were dehydrated using 30% sucrose, embedded in OCT, and sectioned at −20 °C. The sections underwent Oil Red O staining and hematoxylin–eosin (H&E) staining. For Oil Red O staining, the sections were fixed in 4% PFA for 10 min, washed with 60% isopropanol, and then stained with Oil Red O solution for 10 min. After staining, the sections were rinsed with 60% isopropanol and distilled water (ddH_2_O), followed by a 2 min incubation in hematoxylin solution (Beyotime, Shanghai, China). For H&E staining, sections were fixed in 95% ethanol for 20 min and washed twice with PBS. They were subsequently stained with hematoxylin solution for 2 min, rinsed with ddH_2_O, and stained with eosin solution (Beyotime, Shanghai, China) for 1 min. All sections were allowed to air dry, sealed with neutral gum, and examined under a microscope. The positive area for Oil Red O and lipid droplets was quantified using ImageJ v1.50d.

### 2.19. RNA-Seq Analysis

RNA-Seq data for hepatocellular carcinoma (HCC) patients were obtained from cBioPortal (Liver Hepatocellular Carcinoma, TCGA Firehose Legacy), as detailed in Appendix A. The samples were categorized into high- and low-expression groups based on the median expression levels of ATF3. Subsequently, the RNA-Seq data underwent gene set enrichment analysis (GSEA) using multiple gene sets, along with a summary analysis of the results.

### 2.20. Gene Expression Analysis

mRNA expression data for normal and hepatocellular carcinoma (HCC) patients were sourced from cBioPortal (Liver Hepatocellular Carcinoma, TCGA PanCancer Atlas), as presented in Appendix A.

### 2.21. Kaplan–Meier (KM) Survival Analysis

The KM plot was conducted on the following website: https://kmplot.com/analysis/, accessed on 19 October 2022 [24].

### 2.22. Correlation Analysis

The correlation analysis of gene expression level in liver cancer was conducted using Correlation AnalyzeR [25]. Data are listed in Supplementary Appendix A.

### 2.23. Statistical Analysis

Gene expression analysis was conducted using the Kruskal–Wallis test. For other assays, results are expressed as the mean ± standard deviation (S.D.) from at least three independent experiments. Statistical comparisons of various groups were conducted with unpaired two-tailed *t*-tests using Prism 9. *p*-values of <0.05 (*), <0.01 (**), <0.001 (***), and <0.0001 (****) were considered statistically significant.

## 3. Results

### 3.1. ATF3 Suppresses the Amplification and Invasion of HCC Cells

The impact of ATF3 on hepatocellular carcinoma (HCC) cell behavior, encompassing proliferation, invasion, migration, and colony formation, was investigated to delineate its role, given its reported dual functions in HCC pathogenesis [23,26]. Lentiviral vectors encoding ATF3 were engineered by incorporating human ATF3 cDNA into a modified lenti-3 × Flag plasmid. Stable ATF3-overexpressing (ATF3-oe) cell lines were constructed by lentiviral transduction of HCC cell lines. Robust elevation of ATF3 mRNA levels, exceeding 15-fold compared to empty vector (EV)-infected cell lines, was confirmed via qRT-PCR (Figure 1A), affirming the successful establishment of stable cell lines. Subsequently, these ATF3-oe cell lines were subjected to MTT, Transwell, scratch, and colony formation assays. Notably, MTT assay results revealed a marked reduction in HCC cell proliferation upon ATF3 overexpression (Figure 1B). Similarly, Transwell and scratch assays demonstrated a significant attenuation of invasion and migration capacities in ATF3-oe cells, respectively (Figure 1C–F). Consistently, colony formation assay data indicated a notable decrease in the clonogenic potential of HCC cells following ATF3 overexpression (Figure 1G,H). These findings verify the suppressive role of ATF3 overexpression in modulating the oncogenic phenotype of HCC cell lines.

To further corroborate the tumor-suppressive role of ATF3 in HCC, we employed an ATF3 short hairpin RNA (shRNA) oligo to knock down ATF3 expression. The shRNA was cloned into the pLV-H1TetO-GFP-Bsd plasmid, packaged into lentivirus, and used to infect HCC cell lines. Given that the vector’s promoter is inducible by doxycycline, we introduced doxycycline 24 h after lentivirus infection to induce stable ATF3-knockdown (ATF3-kd) cell lines. Concurrently, infected cell lines were maintained without doxycycline as controls. Both control and ATF3-kd cell lines underwent qRT-PCR analysis to ascertain the effectiveness of knockdown. Results indicated a substantial reduction of ATF3 mRNA levels, by approximately 40% in the Huh7 ATF3-kd cell line and 50% in the PLC/PRF/5 ATF3-kd cell line (Figure 1I), affirming the efficacy of ATF3 knockdown. Following successful knockdown, ATF3-kd cell lines were subjected to the same functional assays as ATF3-oe cell lines. MTT assay results unveiled a significant enhancement in HCC cell proliferation upon ATF3 knockdown (Figure 1J). Transwell and scratch assays demonstrated an augmented invasion and migration capacity in ATF3-kd cell lines, respectively (Figure 1K–N), while colony formation assays revealed an increased colony formation ability in these cells (Figure 1O,P). By integrating the outcomes of phenotype assays conducted on both ATF3-overexpressing and ATF3-knockdown HCC cell lines, we provide compelling evidence supporting the suppressive role of ATF3 in HCC tumorigenesis in vitro.

### 3.2. ATF3 Is Correlated with Better Prognosis and Downregulated Lipid Synthesis in HCC

After establishing the suppressive impact of ATF3 on the malignant traits of HCC cells in vitro, we delved deeper into the association between ATF3 expression and the development of HCC by comparing its expression levels in normal and HCC samples. To this end, we analyzed ATF3 mRNA expression data from normal samples (*n* = 50) and HCC samples (*n* = 316) sourced from TCGA database. Our analysis unveiled a significant downregulation of ATF3 mRNA levels in HCC samples compared to normal samples, suggesting a potential association between ATF3 downregulation and the onset and progression of HCC (Figure 2A).

Moreover, we conducted Kaplan–Meier survival analysis using an online tool developed by Gyorffy B [24]. HCC patients were stratified into two groups according to their ATF3 mRNA expression levels. Survival curves were plotted and compared between the two patient cohorts. Our analysis revealed that the high ATF3 expression group exhibited a higher probability of survival compared to the low ATF3 expression group within 80 months. However, in the period spanning 80–120 months, the survival probability of the high ATF3 expression group showed a slight decline, likely attributable to the larger sample size within this timeframe (Figure 2B). Overall, these results corroborate the tumor-suppressive role of ATF3, implying that HCC patients with high ATF3 expression levels have a more favorable prognosis.

To elucidate the cellular events and signaling pathways influenced by ATF3 expression in HCC patients, we accessed the RNA-Seq data of patients from TCGA database. These patients were categorized into two groups based on their ATF3 expression levels. Subsequently, we conducted gene set enrichment analysis (GSEA) on the data. Our analysis, summarizing multiple GSEA gene sets including KEGG gene sets, gene ontology biological process (GOBP) gene sets, and hallmark gene sets, revealed a consistent pattern: the high ATF3 expression group exhibited an overall decrease in lipid metabolism processes compared to the low ATF3 expression group (Figure 2C). Further scrutiny through individual gene set analyses within the KEGG, GOBP, and hallmark gene sets corroborated this observation, indicating a downregulation of fatty acid metabolism in high ATF3 expression patients (Figure 2D). Notably, the GOBP gene set analysis unveiled a downregulation in both fatty acid catabolism and lipid synthesis in the high ATF3 expression group.

Given the well-documented propensity of cancer cells to upregulate de novo fatty acid synthesis to fuel their uncontrolled growth [9,27], and considering the pivotal role of fatty acids as substrates for various lipid species essential for cancer progression, including diglycerides, triglycerides, and phosphoglycerides [28], our findings suggest a compelling association between high ATF3 expression, attenuated lipid biosynthesis, and improved prognosis in HCC patients.

### 3.3. ATF3 Inhibits Lipid Biosynthesis in HCC Cells

To further corroborate the findings from RNA-Seq analysis of HCC patients, we conducted qRT-PCR to assess the expression levels of a panel of lipid metabolism-related genes in both ATF3-overexpressing (ATF3-oe) and ATF3-knockdown (ATF3-kd) HCC cell lines. Overexpressing (ATF3-oe) and ATF3-knockdown (ATF3-kd) HCC cell lines. This panel encompassed genes encoding key enzymes involved in fatty acid synthesis (FASN, ACLY, and SCD) [28], components of lipoproteins (APOA1, APOB, and APOC3) [29,30,31], and regulators of lipid synthesis (mTORC1 and SREBF2) [32,33]. In ATF3-oe Huh7 cell lines, a significant decrease in the expression levels of all examined genes except mTORC1 was observed (Figure 3A). Conversely, in ATF3-kd Huh7 cell lines, the expression levels of FASN, ACLY, SCD, APOB, and APOC3 were elevated, while mTOR expression remained unchanged (Figure 3B). These findings suggest that ATF3 represses lipid synthesis and transportation in HCC cells.

Subsequently, we assessed the lipid accumulation levels in ATF3-oe and ATF3-kd HCC cell lines using Nile Red staining. Consistently, ATF3-oe cell lines exhibited reduced lipid accumulation, while ATF3-kd cell lines showed elevated lipid accumulation (Figure 3C–F). Since lipid de novo synthesis seems to have a direct impact on lipid accumulation in HCC cells, we examined the correlation between ATF3 expression and the key lipogenesis genes (FASN, ACLY, and SCD) in HCC patients. The results revealed a negative correlation between ATF3 expression and the expression of FASN, ACLY, and SCD, consistent with our RNA-Seq and qRT-PCR findings (Figure 3G).

Furthermore, we quantified triglyceride levels in both ATF3-oe and ATF3-kd cell lines, demonstrating that ATF3 overexpression significantly reduces cellular triglyceride levels, whereas ATF3 knockdown leads to increased triglyceride levels (Figure 3H). These results underscore the role of ATF3 in suppressing lipid anabolism by inhibiting the expression of key enzymes in lipogenesis in HCC cells.

### 3.4. ATF3 Represses Lipid Biosynthesis by Inhibiting mTORC1 Signaling

To elucidate the mechanism underlying ATF3-mediated suppression of lipid synthesis, we investigated the signaling pathways related to lipid synthesis that could be modulated by ATF3. Given previous findings indicating that activated mTORC1-S6K phosphorylates SRPK2, leading to enhanced splicing of FASN, ACLY, and SCD mRNA [34], the relationship between ATF3 and mTORC1 signaling is pertinent to our observation of ATF3-mediated downregulation of FASN, ACLY, and SCD mRNA levels. Thus, we hypothesized that mTORC1 might be a potential signaling pathway influenced by ATF3.

Although the mRNA levels of mTORC1 were not obviously changed in either ATF3-oe or ATF3-kd HCC cell lines, mTORC1 functions predominantly through the phosphorylation of S6K to exert its downstream effects [13,34]. Therefore, we assessed the levels of mTOR (a core component of the mTORC1 complex) and phosphorylated S6K (pS6K) in both ATF3-oe and ATF3-kd HCC cell lines via Western blot analysis. Remarkably, pS6K levels were markedly decreased in ATF3-oe cell lines and conversely elevated in ATF3-kd cell lines, whereas mTOR levels remained unaltered (Figure 4A,B). Additionally, we adopted a well-defined molecular biosensor, TORCAR, to measure the mTORC1 activity in single living cells [35]. Since activated mTORC1 kinase leads to the conformational change in TORCAR and results in an increased CFP/YFP ratio, the results further proved that ATF3 downregulates mTORC1 activity in HCC cells, while ATF3 knockdown exhibits the opposite effect (Figure 4C–F). These findings suggest that ATF3-mediated inhibition of mTORC1 signaling is involved in the repression of lipid anabolism.

To further corroborate the impact of mTORC1 as the main signaling pathway in ATF3-mediated suppression of lipid biosynthesis, we treated ATF3-kd HCC cell lines with rapamycin, an inhibitor of mTORC1 [36]. Remarkably, while ATF3 knockdown increased cellular lipid levels, inhibition of mTORC1 reversed the elevated lipid levels induced by ATF3 knockdown (Figure 4G), as confirmed by Nile Red staining (Figure 4H,I). Additionally, quantification of triglyceride (TG) levels in ATF3-kd cell lines with or without rapamycin treatment revealed that mTORC1 inactivation abolished the upregulation of triglycerides caused by ATF3 knockdown in HCC cells (Figure 4J). Free fatty acid (FFA) quantification indicated that while ATF3 alone can reduce FFA synthesis like rapamycin, it strikingly increases the sensitivity of HCC cells to rapamycin (Figure 4K). The inhibitory effect of rapamycin on lipid-synthesis-related genes was confirmed by qRT-PCR. Notably, rapamycin leads to a decreased ATF3 expression level, which is probably a negative feedback regulatory mechanism of HCC cells to maintain necessary lipid synthesis under the stress of rapamycin (Figure 4L). Collectively, these results provide compelling evidence that ATF3 inhibits lipid synthesis by repressing mTORC1 signaling.

### 3.5. SLC7A7 Is Involved in the Inhibition of ATF3 on mTORC1 Signaling

To delineate the mechanism by which ATF3 inhibits mTORC1 signaling in HCC, we turned our attention to the role of amino acids, particularly glutamine, leucine, and arginine, as activators of mTORC1 signaling. These amino acids induce lysosomal translocation of mTORC1, thereby activating its signaling cascade [13]. Given that the SLC3 and SLC7 families are known amino acid transporters, and SLC7A5 has been implicated in mediating leucine influx and mTOR signaling activation, we sought to identify other potential members of the amino acid transporter family that could influence mTORC1 signaling or lipid synthesis [37,38].

Previous studies have implicated SLC7A7 in the activation of mTOR signaling and the regulation of left-right asymmetry in medaka [39], while loss of SLC7A10 has been associated with lipid accumulation in zebrafish [40]. Therefore, we assessed the mRNA expression levels of SLC7A7 and SLC7A10 in ATF3-oe Huh7 cell lines. Strikingly, ATF3 overexpression led to a 2-fold upregulation of SLC7A7 expression, while SLC7A10 expression remained unchanged. Subsequent analysis in ATF3-oe PLC/PRF/5 cell lines revealed a similar trend (Figure 5A). Conversely, in ATF3-kd HCC cell lines, SLC7A7 expression was significantly decreased (Figure 5B). These results collectively suggest that ATF3 upregulation promotes SLC7A7 expression, implicating it as a potential mediator of ATF3′s inhibitory impact on mTORC1 signaling.

To ascertain whether SLC7A7 contributes to ATF3-induced downregulation of lipid synthesis, we generated Huh7 SLC7A7-kd cell lines using the same approach as for ATF3-kd cell lines and assessed the expression of genes related to lipid anabolism and transportation. The results confirmed the efficacy of SLC7A7 shRNA and revealed that, akin to ATF3 knockdown, SLC7A7 knockdown led to increased expression of FASN, ACLY, SCD, APOA1, APOB, and APOC3 (Figure 5C). Notably, SLC7A7 knockdown did not affect ATF3 expression, suggesting that SLC7A7 could be a downstream effector of ATF3, negatively regulating lipid biosynthesis. Correlation analysis between SLC7A7 and ATF3, FASN, ACLY, and SCD further supported these findings, showing a positive correlation with ATF3 and negative correlations with FASN, ACLY, and SCD, consistent with the qRT-PCR results (Figure 5D).

We then explored whether SLC7A7 knockdown elevated lipid synthesis by influencing mTORC1 signaling in SLC7A7-kd HCC cell lines. The results revealed that while SLC7A7 knockdown activated mTORC1, mirroring the effect of ATF3 knockdown, it did not change mTOR expression (Figure 5E). To further substantiate that SLC7A7 is a downstream inhibitor of mTORC1 and lipid synthesis regulated by ATF3, we infected Huh7 SLC7A7-kd cell lines with either empty vector (EV) or ATF3 lentivirus for 24 h and assessed the lipid accumulation level. The findings demonstrated that while ATF3 alone suppressed lipid accumulation, SLC7A7 knockdown reversed this inhibitory effect (Figure 5F,G). Similarly, Western blot assays on EV- or ATF3-infected Huh7 and PLC/PRF/5 cell lines revealed that ATF3 inhibited mTORC1 activation, while SLC7A7 knockdown reversed ATF3-induced mTORC1 inactivation (Figure 5H). Both TG and FFA quantification results indicated that ATF3 represses FFA and TG synthesis, which can be reversed by SLC7A7 knockdown, consistent with Nile Red staining results (Figure 5I,J). Overall, these results underscore the role of SLC7A7 in ATF3-mediated inhibition of mTORC1 signaling and lipid biosynthesis.

### 3.6. ATF3 Binds to SLC7A7 Enhancer and Activates Its Transcription

Next, we intend to disclose the mechanism of how ATF3 upregulates SLC7A7 expression. Since ATF3 is a transcription factor which can both activate and suppress gene transcription [41], we assume that ATF3 could bind to the promoter or enhancer of SLC7A7 to promote its transcription. Utilizing ATF3 ChIP-Seq data from the Encode database (ENCSR402ZCY) in the HepG2 cell line, we identified a peak approximately 3000 bp upstream of the SLC7A7 promoter, suggesting ATF3 binding to this region (Figure 6A). Sequence analysis revealed the presence of a known ATF3-binding site (TTGCATCA) within this region (Figure 6B) [18]. Thus, we hypothesized this region as a potential enhancer of SLC7A7 and performed ChIP-qPCR assays in the Huh7 ATF3-oe cell line to confirm ATF3 binding. We included the ATF3 promoter as a positive control, as ATF3 can bind to and inhibit its own promoter [42]. The results validated ATF3 binding to the potential enhancer of SLC7A7 (Figure 6C,D).

To further validate this region as the enhancer of SLC7A7, we cloned its sequence into a pGL3-promoter vector (Figure 6E), alongside the ATF3 promoter sequence as a control. We co-transfected these plasmids with a pRL-TK plasmid into Huh7 cell lines infected with either EV or ATF3 lentivirus and conducted luciferase reporter assays after 24 h. The findings demonstrated that ATF3 bound to the SLC7A7 potential enhancer and promoted luciferase transcription, while also repressing its own promoter, consistent with prior research (Figure 6F).

Given that H3K27ac and H3K4me1 are known histone modifications of enhancers [43], we analyzed H3K27ac and H3K4me1 histone ChIP-Seq data from the Encode database (ENCSR678LND and ENCSR642HII). Both modifications exhibited peaks at the same position as the potential SLC7A7 enhancer (Figure 6G). ChIP-qPCR assays further confirmed the enrichment of H3K27ac and H3K4me1 on the potential SLC7A7 enhancer (Figure 6H).

In summary, these findings reveal a newly characterized SLC7A7 enhancer upstream of its promoter, flanking the ATF3-binding site. ATF3 binds to this enhancer to activate SLC7A7 transcription, thereby inhibiting lipid synthesis through the ATF3-SLC7A7-mTORC1 axis.

### 3.7. ATF3 Represses Tumor Growth and Lipogenesis In Vivo

Building on our in vitro findings that ATF3 suppresses lipid synthesis and tumorigenesis, we sought to confirm these effects in vivo. We subcutaneously injected the ATF3-kd Huh7 cell line into the flanks of nude mice, dividing them into two groups. The control group received water containing 5% sucrose, while the experimental group was administered water with 5% sucrose and 2 mg/mL doxycycline to induce ATF3 shRNA expression. Tumor growth was monitored, revealing a significant increase in both tumor volume and weight in the experimental group compared to controls (Figure 7A–C), suggesting that ATF3 knockdown promotes tumor growth in HCC, consistent with our in vitro results.

To investigate whether SLC7A7 is involved in this effect, we detected its expression in tumor tissues. Given the lack of a sensitive SLC7A7 antibody for immunohistochemistry, we analyzed tumor lysates via Western blotting. Results showed that ATF3 knockdown reduced SLC7A7 expression, activated mTORC1 signaling, and upregulated lipogenesis-related enzymes (Figure 7D), implying that downregulation of SLC7A7 may contribute to the enhanced tumor growth observed upon ATF3 knockdown. This hypothesis was further supported by Transwell and MTT assays, where HCC cell lines overexpressing both ATF3 and SLC7A7 exhibited greater suppression of invasion and proliferation compared to cells overexpressing ATF3 alone (Figure 7E–G). Collectively, these findings demonstrate that ATF3 inhibits tumorigenesis in vivo through regulation of SLC7A7 expression.

We next examined the impact of ATF3 on lipid synthesis in vivo. Oil Red O staining of tumor tissues revealed a significant increase in lipid deposition in the ATF3-knockdown group compared to controls (Figure 7H,I). Furthermore, H&E staining indicated that tumors with ATF3 knockdown exhibited more cytoplasmic vacuoles, attributable to the increased lipid droplets (Figure 7J,K). These results indicate that ATF3 represses lipogenesis in HCC in vivo.

## 4. Discussion

Tumor cells exhibit a voracious demand for energy and biomass to sustain their relentless proliferation. Metabolic reprogramming stands out as a hallmark of these cells, enabling them to generate the necessary energy and intermediates for biomass production [44]. In addition to the well-known reprogramming of glucose metabolism, characterized by the Warburg effect, recent attention has also turned towards the reprogramming of lipid metabolism in cancer [9,27]. Lipids serve as both essential components of cellular and organelle membranes and key signaling molecules within cells. In cancer cells, lipogenesis is upregulated, resulting in the deposition of excessive lipids stored as lipid droplets, which can be metabolized via β-oxidation to produce ATP [45,46,47]. Investigating pathways related to the reprogramming of tumor lipid metabolism holds promise for identifying new therapeutic targets and advancing our theoretical understanding of cancer treatment.

HCC stands out as a leading cause of malignancy-related mortality. Despite advancements in treatment, approximately 70% of late-stage HCC patients face the grim reality of recurrence and metastasis following surgical resection [48]. The urgent need for effective therapies to combat HCC is underscored by its poor prognosis. Notably, enzymes involved in lipogenesis are dysregulated in HCC, highlighting elevated lipogenesis as a potential therapeutic target [49]. However, the precise association between HCC development and lipid synthesis pathways remains incompletely understood.

ATF3, a transcription factor participating in mediating various biological processes including metabolism, immunity, and tumorigenesis, presents an intriguing subject of study. Despite its known roles in regulating both oncogenesis and lipogenesis, the precise contribution of ATF3 to HCC development remains contentious [18]. Moreover, the impact of ATF3 on lipid metabolism in HCC has not been investigated. Therefore, our study seeks to elucidate the role of ATF3 in HCC progression and lipid metabolism.

Through functional assays conducted in both ATF3-overexpressing (ATF3-oe) and ATF3-knockdown (ATF3-kd) cell lines, we verified the tumor suppressor role of ATF3 in HCC development. Transcriptome analysis of HCC patients revealed that high ATF3 expression affects lipid metabolism by downregulating lipid anabolism, a finding corroborated by experimental evidence. We discovered that ATF3 inhibits lipogenesis in HCC cells by suppressing mTORC1 signaling, with SLC7A7 playing a crucial role in this process. ATF3 binds to the enhancer region of SLC7A7, activating its transcription, which, in turn, inhibits mTORC1 activation. Our findings unveil the ATF3-SLC7A7-mTORC1 axis as a critical regulator of lipid synthesis and HCC development (Figure 8).

The limitations of this study present avenues for future investigation. First, while we demonstrate that ATF3 binds to and activates the SLC7A7 enhancer and identify its binding sequence, previous studies have shown that ATF3 interacts with various co-factors, such as p53, JunB, and HDAC1, to mediate transcriptional regulation [50,51]. To gain a comprehensive understanding of the ATF3-SLC7A7 pathway, mass spectrometry could be employed to identify potential co-factors involved in this interaction. Such insights would illuminate the broader regulatory network orchestrated by ATF3. Second, we speculate on the potential dual role of SLC7A7, a transmembrane arginine–leucine transporter, in mTORC1 activation, given the role of leucine and arginine as mTORC1 activators [13,52,53]. While our study reveals an inhibitory role for SLC7A7 in mTORC1 signaling in HCC, further investigation into its structure and function, including measurements of arginine efflux and leucine influx, is warranted. Future research addressing these aspects is expected to refine our understanding of the ATF3-SLC7A7 enhancer interaction and the complex regulatory dynamics of SLC7A7 in mTORC1 signaling.

To advance our findings toward clinical application, several key areas warrant further investigation. First, developing small molecules or gene therapy approaches that modulate ATF3 activity or mimic its effects on SLC7A7 could serve as promising therapeutic strategies. High-throughput screening for ATF3 activators and the design of enhancer-targeted delivery systems for ATF3 transactivation are potential avenues to explore. Second, patient-derived organoids and single-cell transcriptomic approaches could be employed to validate the ATF3-SLC7A7-mTORC1 axis across different HCC subtypes, facilitating the stratification of patients who might benefit from targeted therapies. By integrating these findings with clinical studies, we aim to bridge the gap between mechanistic discoveries and therapeutic applications, potentially contributing to the development of novel treatments for HCC.

## Figures and Tables

**Figure 1 cells-14-00253-f001:**
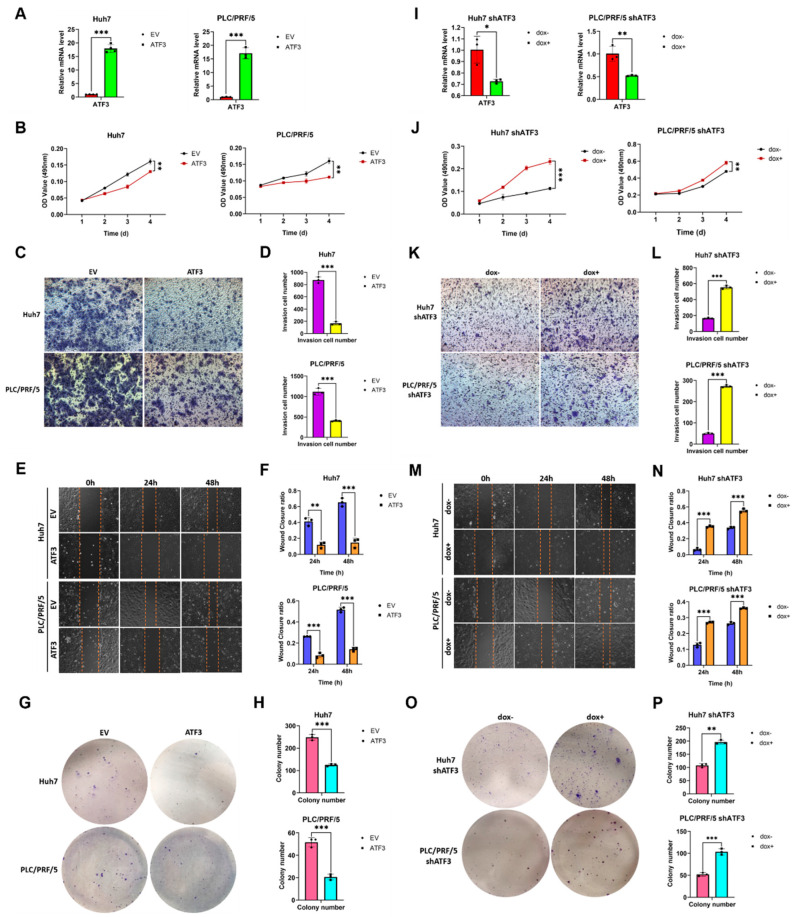
**ATF3 acts as a tumor suppressor in HCC.** (**A**) qRT-PCR analysis of ATF3-overexpressing Huh7 and PLC/PRF/5 cell lines. (**B**) MTT assay of cell lines above. (**C**) Transwell assay of cell lines above. (**D**) Column of Transwell assay result. (**E**) Scratch assay of cell lines above. (**F**) Column chart of scratch assay result. (**G**) Colony formation assay of cell lines above. (**H**) Column chart of colony formation assay result. (**I**) qRT-PCR analysis of ATF3-knockdown Huh7 and PLC/PRF/5 cell line. (**J**) MTT assay of cell lines above. (**K**) Transwell assay of cell lines above. (**L**) Column chart of Transwell assay result. (**M**) Scratch assay of cell lines above. (**N**) Column chart of scratch assay result. (**O**) Colony formation assay of cell lines above. (**P**) Column chart of colony formation assay result. * denotes *p* < 0.05, ** denotes *p* < 0.01, and *** denotes *p* < 0.001.

**Figure 2 cells-14-00253-f002:**
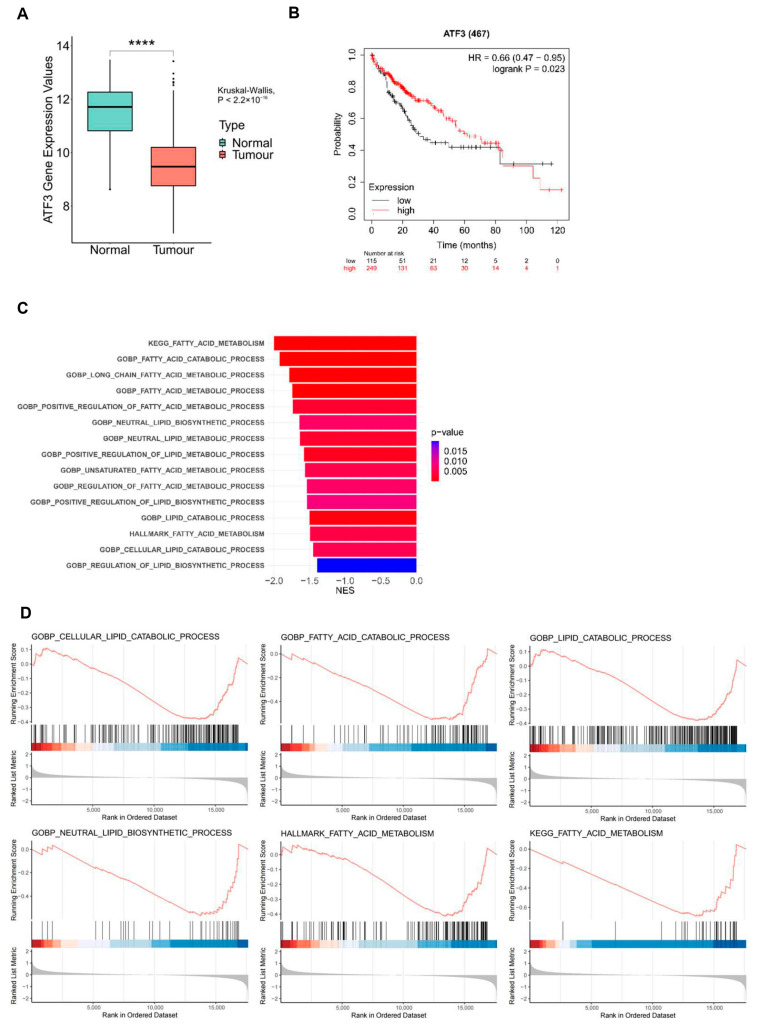
**ATF3 is associated with good prognosis and downregulated lipid anabolism in HCC.** (**A**) Gene expression analysis of ATF3 in normal patients and HCC patients. (**B**) Kaplan–Meier survival analysis of HCC patients with high ATF3 expression and low ATF3 expression. (**C**) Summary of GSEA analysis of RNA-Seq results of patients with high ATF3 expression compared to patients with low ATF3 expression. (**D**) Individual GSEA analysis results of KEGG, GOBP, and hallmark gene sets. **** denotes *p* < 0.0001.

**Figure 3 cells-14-00253-f003:**
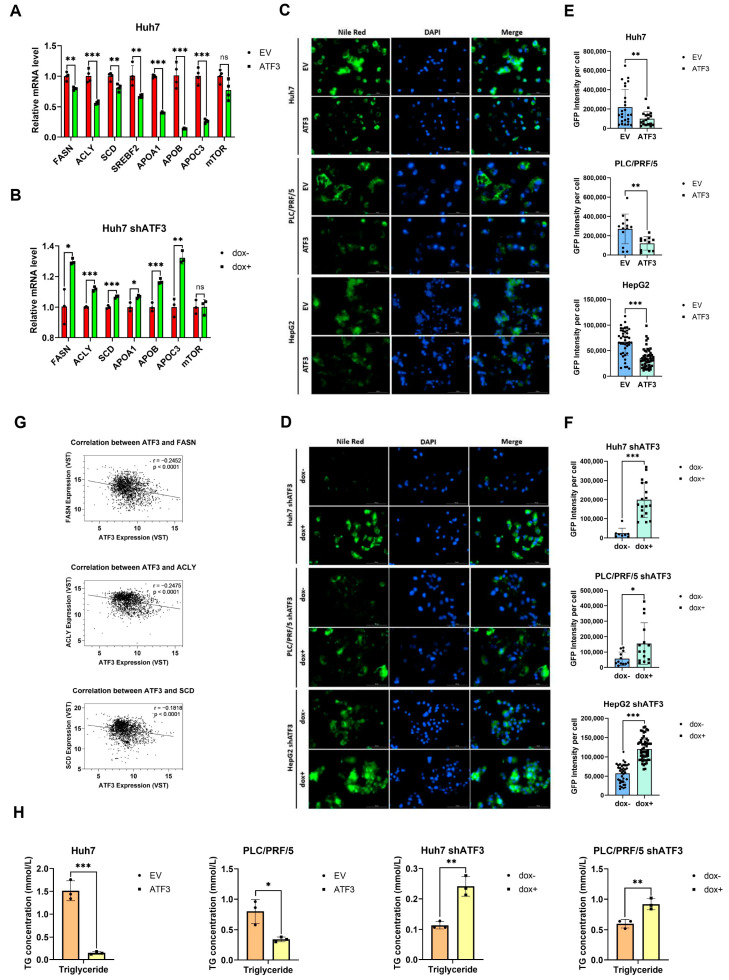
**ATF3 suppresses lipid synthesis in HCC.** (**A**,**B**) qRT-PCR analysis of lipid-related genes in ATF3-overexpressing Huh7 cell line and ATF3-knockdown Huh7 cell line. (**C**) Nile Red staining of ATF3-overexpressing Huh7, PLC/PRF/5, and HepG2 cell lines. (**D**) Nile Red staining of ATF3 knockdown in above cell lines. (**E**,**F**) Fluorescence intensity quantification of Nile Red staining assay. (**G**) Correlation analysis between ATF3 and FASN, ACLY and SCD expression in Correlation AnalyzeR. (**H**) Triglyceride quantification of ATF3-overexpressing and ATF3-knockdown Huh7 and PLC/PRF/5 cell lines. * denotes *p* < 0.05, ** denotes *p* < 0.01, and *** denotes *p* < 0.001.

**Figure 4 cells-14-00253-f004:**
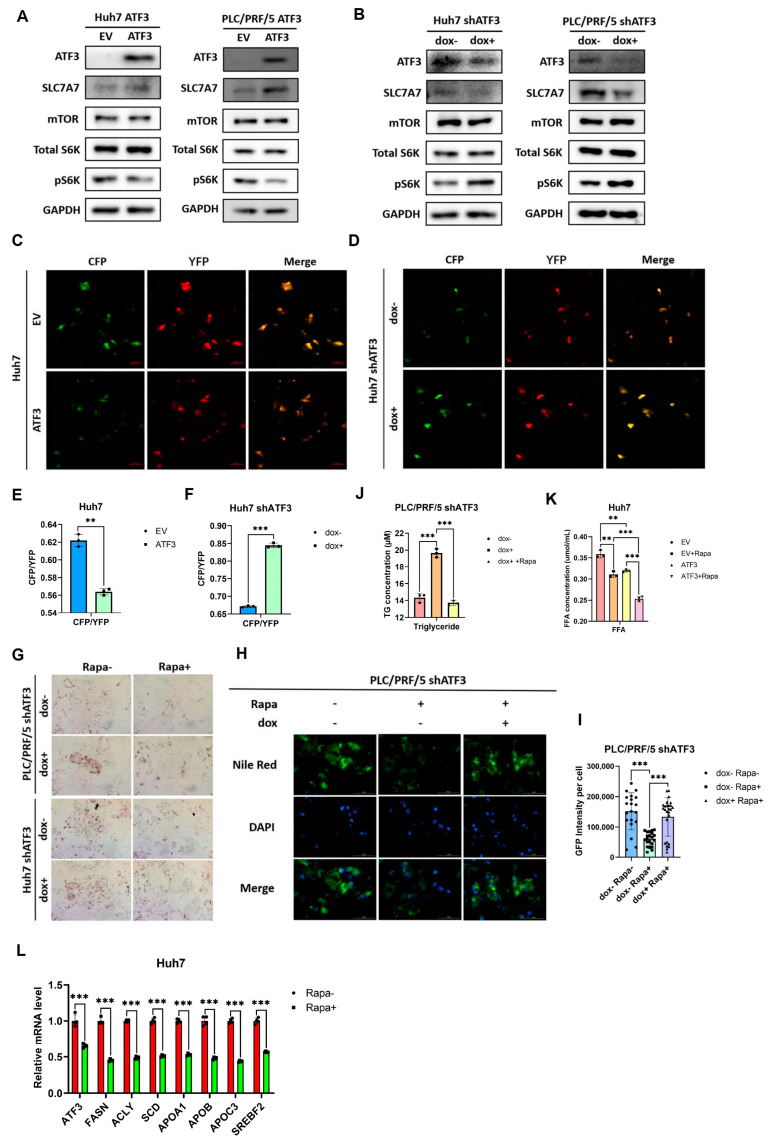
**ATF3 inhibits mTOR signaling to repress lipid synthesis.** (**A**) Western blot of ATF3-overexpressing Huh7 and PLC/PRF/5 cell lines. (**B**) Western blot of ATF3-knockdown Huh7 and PLC/PRF/5 cell lines. (**C**,**D**) Confocal assay of ATF3-oe and ATF3-knockdown Huh7 cell line transfected with TORCAR. (**E**,**F**) CFP/YFP ratio of confocal assay. (**G**) Oil Red O staining assay of ATF3-knockdown Huh7 and PLC/PRF/5 cell lines with or without rapamycin. (**H**) Nile Red staining assay of ATF3-knockdown PLC/PRF/5 cell line with or without rapamycin. (**I**) Fluorescence intensity quantification of Nile Red Staining assay. (**J**) Triglyceride quantification of ATF3-knockdown PLC/PRF/5 cell line with or without rapamycin. (**K**) FFA quantification of ATF3-oe Huh7 cell line with or without rapamycin. (**L**) qRT-PCR analysis of Huh7 cell line with or without rapamycin. ** denotes *p* < 0.01, and *** denotes *p* < 0.001.

**Figure 5 cells-14-00253-f005:**
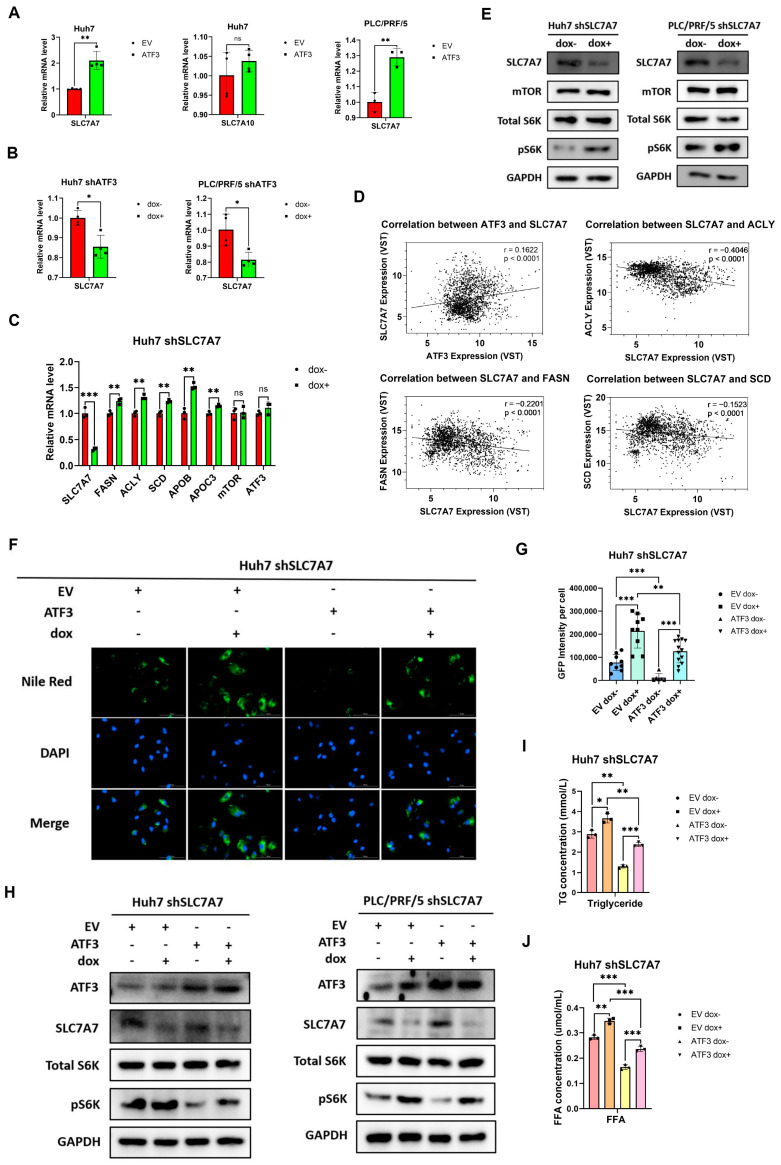
**ATF3 upregulates SLC7A7 to influence mTOR signaling and lipid synthesis.** (**A**) qRT-PCR analysis of SLC7A7 in ATF3-overexpressing Huh7 and PLC/PRF/5 cell lines, and SLC7A10 in ATF3-overexpressing Huh7 cell line. (**B**) qRT-PCR analysis of SLC7A7 in ATF3-knockdown Huh7 and PLC/PRF/5 cell lines. (**C**) qRT-PCR analysis of lipid-related genes in SLC7A7-knockdown Huh7 cell line. (**D**) Correlation analysis between SLC7A7 and ATF3, FASN, ACLY, and SCD expression in Correlation AnalyzeR. (**E**) Western blot of SLC7A7-knockdown Huh7 and PLC/PRF/5 cell lines. (**F**) Nile Red staining assay of SLC7A7-knockdown Huh7 cell line with EV or ATF3 overexpression. (**G**) Fluorescence intensity quantification of Nile Red Staining assay. (**H**) Western blot of SLC7A7-knockdown Huh7 and PLC/PRF/5 cell lines with EV or ATF3 overexpression. (**I**,**J**) Triglyceride and FFA quantification of SLC7A7-knockdown Huh7 cell line with EV or ATF3 overexpression. * denotes *p* < 0.05, ** denotes *p* < 0.01, and *** denotes *p* < 0.001.

**Figure 6 cells-14-00253-f006:**
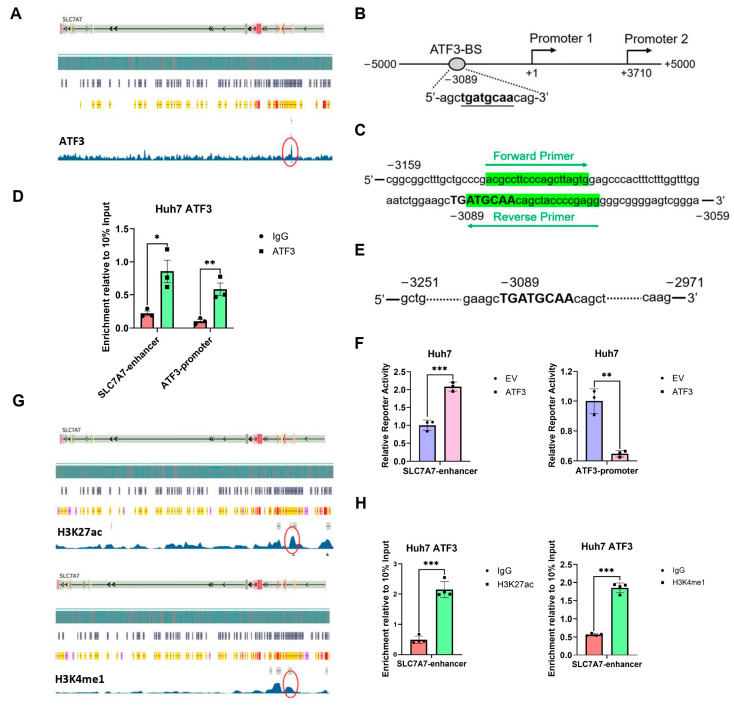
**ATF3 interacts with SLC7A7 enhancer to upregulate its expression.** (**A**) Transcription factor ATF3-targeted ChIP-seq analysis of HepG2 cell line from Encode. (**B**) Schematic figure of potential enhancer sequence and ATF3-binding motif. (**C**) Schematic figure of primers in ChIP-qPCR. (**D**) ChIP-qPCR analysis of ATF3 enrichment on SLC7A7 potential enhancer and ATF3 promoter in ATF3-overexpressing Huh7 cell line. ATF3 promoter serves as positive control. (**E**) Schematic figure of SLC7A7 enhancer sequence cloned into pGL3 promoter. (**F**) Luciferase reporter assay of SLC7A7 enhancer and ATF3 promoter basing on pGL3 promoter in ATF3-overexpressing Huh7 cell line. (**G**) Histone H3K27ac and H3K4me1-targeted ChIP-seq analysis of liver tissue from Encode. (**H**) ChIP-qPCR analysis of H3K27ac and H3K4me1 enrichment on SLC7A7 enhancer in ATF3-overexpressing Huh7 cell line. The red circle represents the signal peak on the SLC7A7 enhancer. * denotes *p* < 0.05, ** denotes *p* < 0.01, and *** denotes *p* < 0.001.

**Figure 7 cells-14-00253-f007:**
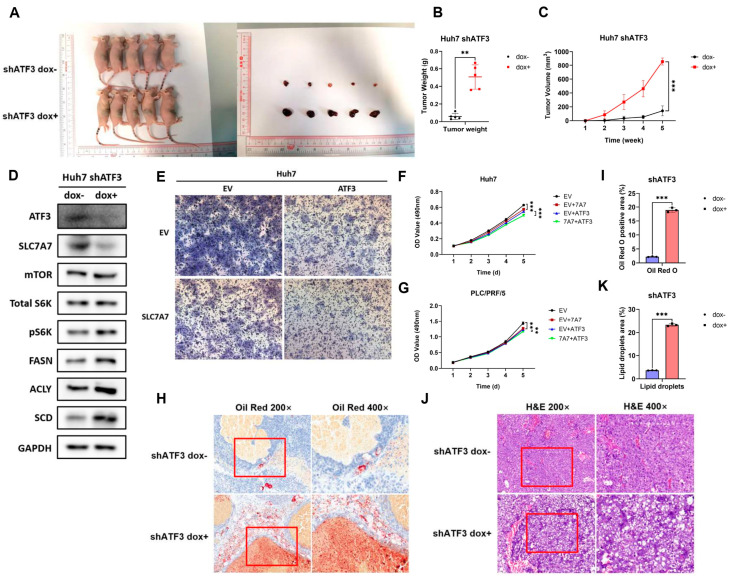
**ATF3 represses tumor growth and lipid synthesis in vivo by regulating SLC7A7 expression.** (**A**) Photographs of mice and tumors in the mouse xenograft assay. (**B**) Tumor weight and (**C**) tumor volume. (**D**) Western blot of SLC7A7 and ATF3 expression level in tumor tissues. (**E**) Transwell assay of ATF3-overexpressing and both ATF3- and SLC7A7-overexpressing Huh7 cell lines. (**F**) MTT assay of cell lines above. (**G**) MTT assay of ATF3-overexpressing and both ATF3- and SLC7A7-overexpressing PLC/PRF/5 cell lines. (**H**) Immunohistochemistry assay of Oil Red O staining. (**I**) Quantification of Oil Red O positive area. (**J**) Immunohistochemistry assay of H&E staining. (**K**) Quantification of lipid droplet area. The red square frame represents the region of 400× magnification under the microscope. ** denotes *p* < 0.01, and *** denotes *p* < 0.001.

**Figure 8 cells-14-00253-f008:**
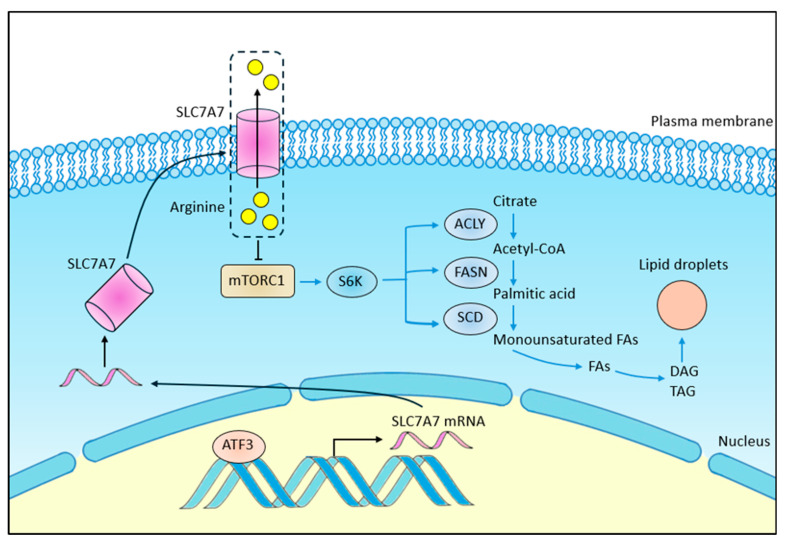
Concept graphic of the link between ATF3-SLC7A7-mTORC1 axis and lipogenesis pathway.

## Data Availability

The original contributions presented in this study are included in the article/Appendix A. Further inquiries can be directed to the corresponding author.

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
