# Peer review of "ATF3-SLC7A7 Axis Regulates mTORC1 Signaling to Suppress Lipogenesis and Tumorigenesis in Hepatocellular Carcinoma"

_cells, 2025, doi:10.3390/cells14040253_

Round 1
Reviewer 1 Report
Comments and Suggestions for Authors
Here, the authors investigate the role of the ATF3-SLC7A7 axis in regulating mTORC1 signaling to suppress lipogenesis and tumorigenesis in hepatocellular carcinoma (HCC). The article is well written, the figures of good quality and the statistics appropriate.
Major Findings:
The authors show that ATF3 suppresses lipogenesis: ATF3 inhibits mTORC1 signaling, reducing lipid synthesis in HCC.
SLC7A7 is a key mediator of this process as ATF3 activates SLC7A7, which in turn restrains mTORC1 activity.
Overexpression of ATF3 in HCC cell lines confirms its role as a tumor SUPPRESSOR. This indicates the therapeutic relevance of targeting this axis in HCC.
Minor Points to Address:
A limitations paragraph can also be added describing for example how further exploration of the exact mechanisms by which ATF3 and SLC7A7 interact.
A long term prospect paragraph describing how Clinical Relevance could be improved by Additional studies that would be needed to translate these findings into therapeutic strategies for HCC.
Reviewer 2 Report
Comments and Suggestions for Authors
The manuscript has a good experimental design and study approach. The results are solid and straightforward, and the manuscript is well-organized and well-written. Therefore, it can be considered for further review after addressing the following minor issues:
- Regarding mTOR expression in Figure 2, the authors examined its mRNA expression. Could they also test its protein expression?
2. How about the effect on the downstream target genes of mTROC1 in the study? Did the author try rescue experiments by overexpressing mTOR while simultaneously overexpressing ATF3 in HCC cell lines to confirm their relationship?
- HepG2 cells are hepatoblastoma cell lines, and many researchers now prefer using other HCC cell lines. Since no experiments were conducted using HepG2 cells in this study, this information could be omitted.
- Please add references to the Methods section, as studies are typically conducted based on prior research.
- In Figure 5, did the authors analyze the correlation between SLC7A7 and mTOR expression?
- In the Figure 7 study, did the authors also examine mTOR signaling and lipid synthesis-related genes, as was done in Figure 5?
